# QVD: Post-training Quantization for Video Diffusion Models

## ABSTRACT

Recently, video diffusion models (VDMs) have garnered significant attention due to their notable advancements in generating coherent and realistic video content. However, processing multiple frame features concurrently, coupled with the considerable model size, results in high latency and extensive memory consumption, hindering their broader application. Post-training quantization (**PTQ**) is an effective technique to reduce memory footprint and improve computational efficiency. Unlike image diffusion, we observe that the temporal features, which are integrated into all frame features, exhibit pronounced skewness. Furthermore, we investigate significant inter-channel disparities and asymmetries in the activation of video diffusion models, resulting in low coverage of quantization levels by individual channels and increasing the challenge of quantization. To address these issues, we introduce the first PTQ strategy tailored for video diffusion models, dubbed **QVD**. Specifically, we propose the **H**igh **T**emporal **D**iscriminability Quantization (**HTDQ**) method, designed for temporal features, which retains the high discriminability of quantized features, providing precise temporal guidance for all video frames. In addition, we present the **S**cattered **C**hannel **R**ange **I**ntegration (**SCRI**) method which aims to improve the coverage of quantization levels across individual channels. Experimental validations across various models, datasets, and bit-width settings demonstrate the effectiveness of our QVD in terms of diverse metrics. In particular, we achieve near-lossless performance degradation on W8A8, outperforming the current methods by 205.12 in FVD.

## CCS CONCEPTS

• **Computing methodologies** → **Artificial intelligence**.

## KEYWORDS

post-training quantization, video diffusion models, multimodal

## 1 INTRODUCTION

The diffusion model has experienced vigorous development in vision generation tasks due to its high controllability, photorealistic generation, and impressive diversity. Recently, research on video tasks based on diffusion models has gained increasing attention, driving the emergence of numerous attractive applications including, but not limited to, text-to-video [5, 18, 20, 65], image-guided video generation [4, 18, 25, 44], video editing [14, 41, 59], and other conditionally guided video generation tasks [53, 54, 60].

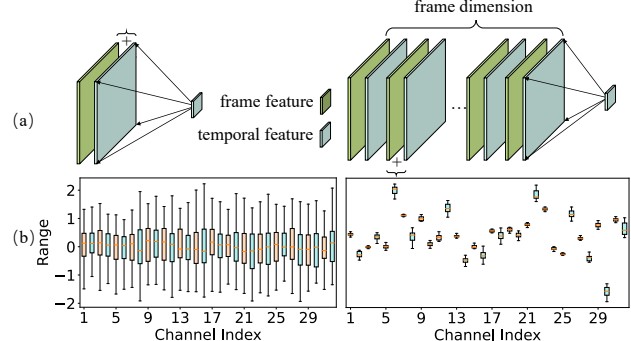

**Figure 1: Comparison of image diffusion (left) and video diffusion (right). (a) Features of all frames rely on the same temporal feature in VDMs. (b) Significantly inter-channel variation issue occurs in temporal attention modules.**

Despite the remarkable effects of diffusion models, their relatively slow inference speed and substantial memory usage have hindered their broader application, particularly in video tasks. The core reasons for these limitations include: 1) The denoising procedure encompasses several hundred iterations, and 2) The extension of frame dimensions results in a notable escalation in memory utilization compared with image diffusion models. There are primarily two strategies to overcome such bottlenecks: minimizing the number of iteration steps, including [37, 51], and optimizing the efficiency of individual denoising operations through techniques like pruning [16, 32], distillation [30, 39], or quantization [33, 47]. The former only focuses on the first issue while ignoring the significant memory consumption. In this work, we mainly study the quantization of video diffusion models.

Model quantization is a widely adopted and practical approach for reducing model footprint and accelerating inference by mapping floating-point values into low-bit integers. Among various quantization methods, post-training quantization (PTQ) requires no retraining or fine-tuning of the model and incurs minimal overhead, making it more practical in deployment. PTQ methods have been extensively studied in image diffusion models [19, 27, 33, 47, 49]. However, these methods exhibit significant performance degradation when directly applied to video diffusion models. We discover that the rationale lies in two aspects, *i.e.*, the introduction of the frame dimensions and temporal attention modules in video diffusion models. Specifically, based on the following two observations, we explore the difficulty of quantization for VDMs:

**Observation 1: Highly reliant on discriminable temporal features.** As illustrated in Figure 1(a), in a single denoising iteration, the features of all frames rely on the same temporal feature, which implies that disturbances arising from the quantization of temporal features will impact the generative quality of all frames. Furthermore, we observe that the uniform quantizer leads to the homogenization of temporal features, demonstrating a significant performance gap

                                                                                                                                          

compared to those utilizing full-precision temporal features. Upon analyzing the distribution of temporal features, we observe a pronounced skewness near zero, where outliers often exceed thousands of times the magnitude of regular values, making conventional uniform quantizers unsuitable for temporal features.

**Observation 2: Inter-channel variations reduce the coverage of quantization levels.** As depicted in Figure 1(b), in the image diffusion model, activation across different channels tends to be concentrated, with the range of activation for individual channels closely approximating the range of overall activation. Each channel covers nearly all quantization levels, indicating low quantization difficulty. In contrast to the image diffusion model, the activation values of the temporal attention module in video diffusion models are discrete and asymmetric across channels. The range of individual channels is significantly narrower than the overall activation. As a result, each channel accesses only a tiny fraction of quantization levels, posing a new challenge for PTQ.

To tackle these obstacles, we propose **QVD**, the first post-training quantization scheme for video diffusion models. Figure 2 shows the overall pipeline of the QVD. To mitigate the problem in observation 1, we introduce the **H**igh **T**emporal **D**iscriminability **Q**uantization (**HTDQ**), which contains a **Hi**gh **Di**scriminability **T**emporal **Q**uantizer (**HiDi-TQ**) to prioritize numerous near-zero values and retains high identifiability of time, and the **T**emporal **D**iscriminability score (**TDScore**) to evaluate the similarity of adjacent temporal features. To settle the issue in observation 2, we introduce a **S**cattered **C**hannel **R**ange **I**ntegration (**SCRI**) method, which employs a per-channel integration operation to enhance the coverage of quantization levels by individual channels, therefore handle the discrete range of activations. We conduct comprehensive experiments to validate the superiority and versatility of our QVD.

In summary, our contributions are as follows:

- We propose **QVD** quantization framework, which, to our knowledge, is the first PTQ method tailored explicitly for video diffusion models.
- We identify the critical inter-channel variations issue in video diffusion models and highlight the significance of accurate and discriminable temporal features for video generation.
- We introduce **SCRI** to improve the coverage of quantization levels across individual channels, **TDScore** to quantify temporal similarity, and the **HiDi-TQ** quantizer to keep high discriminability of temporal features.
- Extensive experiments on various models and datasets demonstrate the superiority of QVD, which results in a 257.9 decrease in the FVD for the W6A8 PTQ of video diffusion models compared to existing methods in the image domain.

## 2 RELATED WORK

### 2.1 Video Diffusion

Recently, Diffusion Probabilistic Models [22, 50] have overtaken Generative Adversarial Networks (GANs) [13] as the leading approach in generative modeling, establishing a new benchmark for the field. Following the success of image diffusion techniques, video diffusion has also received widespread interest. VDM [23] becomes the pioneer in video generation and adopts 3D U-Net [12] structure. Some text-to-video (T2V) methods, such as MagicVideo [64],

LVDM [21] utilize the Latent Diffusion Model (LDM) [46] and plug temporal modeling technique to it. Subsequent T2V schemes [5, 55, 55, 62] extend the single-stage to multi-stages. Image-to-video (I2V) methods, as another promising scheme, generate the video from a conditional image. Initially, LaMD [26] focuses on training an autoencoder to isolate motion information contained in videos. Stable Video Diffusion leverages text-to-image pretraining, video pretraining, and high-quality video finetuning to produce high-resolution videos. AnimateDiff [18] integrates the LoRA [24] and avoids the time-consuming retraining. Other conditions, such as pose [29, 38], motion [7, 61], sound [31, 36] are also proposed. Substantial efficient solutions, including retraining-free sampler [2, 35, 37] and retraining-based methods [51, 63]. The first aims to decrease the number of sampling steps and the second is time-consuming. However, there is a gap in the research on video diffusion, and our work is the first to undertake a study in this area.

### 2.2 Quantization

Quantization has achieved substantial advancements in the domain of neural network acceleration, as corroborated by numerous scholarly investigations [6, 10, 28, 34, 40, 42, 56, 58]. Mainstream quantization schemes can be briefly classified into two categories: quantization-aware training (QAT) [9, 15] and post-training quantization (PTQ) [6, 10, 40]. QAT aims to retrain the network on the whole dataset, while PTQ only requires a small amount of unlabeled datasets for calibration. Several classical quantization methods, such as MinMax [28], Percentile [58], LSQ [15], PACT [9] are proposed successively for the convolutional neural networks. In recent years, the quantization of diffusion models [19, 27, 33, 47] has garnered widespread attention within the academic community. PTQ4DM [47] first discovers the difficulty of multiple-step activation distribution and generates the calibration data from a kew-normal distribution. Q-diffusion [33] introduces the uniform sampling calibration and split shortcut quantization for the bimodal activation distribution of the shortcut layers. PTQD [19] decomposes the quantization noise into interrelated and residual parts. TFMQ-DM [27] addresses the temporal feature disturbance and optimizes them separately. However, these existing methods mainly focus on image diffusion. In comparison, video diffusion necessitates significantly greater computational resources and storage. To the best of our knowledge, our work is the first to conduct the quantization for video diffusion models.

## 3 PRELIMINARIES

### 3.1 Diffusion Models

Diffusion models employ a sophisticated approach to image generation, relying on the application of Gaussian noise through a Markov chain in a forward process and a learned reverse process to generate high-quality images. Beginning with an initial data sample $x_0 \sim q(x)$ from a real distribution $q(x)$, the forward diffusion process incrementally adds Gaussian noise over $T$ steps:

$$q\left(x_t | x_{t-1}\right) = \mathcal{N}\left(x_t; \sqrt{1 - \beta_t} x_{t-1}, \beta_t I\right), \qquad (1)$$

where $t$ is an arbitrary timestep and $\{\beta_t\}$ is the variance schedule. The reverse process, in contrast, aims to denoise the Gaussian noise $x_T \sim \mathcal{N}(0, I)$ into the target distribution by estimating $q\left(x_{t-1} | x_t\right)$. In

every step of the reverse process, marked by $t$, the model estimates the conditional probability distribution using a network $\epsilon_\theta(\mathbf{x}_t, t)$, which incorporates both the timestep $t$ and the prior output $\mathbf{x}_t$ as its inputs:

$$x_{t-1} \sim p_\theta(x_{t-1}|x_t) =$$
$$\mathcal{N}\left(x_{t-1}; \frac{1}{\sqrt{\alpha_t}}\left(\mathbf{x}_t - \frac{1-\alpha_t}{\sqrt{1-\bar{\alpha}_t}}\epsilon_\theta(\mathbf{x}_t, t)\right), \beta_t \mathbf{I}\right), \quad (2)$$

where $\alpha_t = 1 - \beta_t$ and $\bar{\alpha}_t = \prod_{i=1}^{T} \alpha_i$. When extending the image diffusion to video diffusion, the latent noise adds a new dimension $K$ which denotes the length of the video frames.

## 3.2 Model Quantization

Model quantization represents a technique for model compression, vital in optimizing neural networks for resource-constrained environments. Quantization transforms the network's weights and activations from a floating-point to a low-bit representation, thereby reducing memory footprint and computational intensity. This transformation is quantitatively described as follows:

$$w_q = \text{clamp}\left(\lfloor\frac{w}{s}\rceil + z, 0, 2^b - 1\right), \quad (3)$$

$$\hat{w} = s \cdot (w_q - z) \approx w, \quad (4)$$

where $w$ and $\hat{w}$ denote the original and de-quantized weights or activations, $w_q$ is the quantized integer representation, $s$ represents the scaling factor, $z$ is the zero point, and $b$ is the bit precision corresponding to $2^b$ quantization levels. The uniform quantization here has equal intervals between each level. Quantization introduces an approximation (quantization) and a subsequent reconstruction (de-quantization) of network parameters.

Expanding upon uniform quantization, our research incorporates logarithmic quantizer that also aligns well with the hardware-oriented aspects. For instance log2 quantization, used primarily on positive activation values, is succinctly represented as:

$$w_q = \text{clamp}\left(\lfloor-\log_2\frac{w}{s}\rceil, 0, 2^b - 1\right), \quad (5)$$

$$\hat{w} = s \cdot 2^{-w_q} \approx w. \quad (6)$$

This method, like uniform quantization, involves the scaling factor $s$ but introduces a logarithmic approach to the quantization process. It offers rapid bit-shifting operations, making it a strategic choice for efficiently implementing models on hardware platforms. The integration of log2 quantization into our model framework further exemplifies our commitment to enhancing computational efficiency while maintaining fidelity in the intricate process of diffusion-based video generation.

## 3.3 Temporal Features in Video Diffusion Models

In the video diffusion model, the time step $t$ is encoded by the function $h(\cdot)$ into a temporal encoding, which is then mapped to temporal features by the embedding function $f(\cdot)$. These temporal features are channel-adjusted by the function $g(\cdot)$ in every Resnet-block3D of the noise estimation network and fused with all frame

features. Formally, for the $i$-th Resnetblock3D, this process can be described using the following equation:

$$\mathbf{F}_t = \mathbf{F} + g_i(f(h(t))), \quad (7)$$

where $\mathbf{F}_t$ represents the frame feature fused with the projected temporal feature. We denote the temporal feature at time-step t as $\mathbf{T}_{emb}^t$:

$$\mathbf{T}_{emb}^t = f(h(t)). \quad (8)$$

## 4 MODIFICATIONS

### 4.1 High Temporal Discriminability Quantization

As previously discussed, video diffusion models introduce a frame dimension, which enables the model to predict the noise for N frame features in each inference, a concept illustrated in Figure 2. In contrast, in image diffusion models, the frame count remains fixed at one. Structurally, temporal features are integrated into the features of each frame, and the quantization noise consequently spreads across all frames. Features from different frames are further fused within the temporal attention blocks, which results in the effects of quantization being compounded. As indicated in Table 4, omitting the quantization of temporal features leads to a significant reduction in the FVD by 160.82 compared to the uniform baseline. These findings highlight the critical importance of precise temporal features for video diffusion models.

To further explore this, we investigate the temporal features to explain why uniform quantizers fail to function effectively. The distribution of temporal features demonstrates a pronounced skewness, with a majority of the values aggregating near zero, and outliers are several orders of magnitude greater than the typical values observed as depicted in Figure 4(a). Even with a 10-bit uniform quantizer, dense intervals utilize only one of the 1024 quantization levels, as depicted in Figure 4(b). This causes most values in the temporal features to collapse to a single value, as shown in Figure 3(b), severely impairing their distinguishability. The log2 quantizer, as shown in Figure 4(c), allocates more quantization levels to dense intervals, preserving the distribution of small values in the temporal features and thus their discriminability. As indicated in Table 4, the log quantizer reduces FVD by 131.63 compared to the linear quantizer, demonstrating its effectiveness. However, we note that despite the improved FVD with the log quantizer, it incurs a greater L2 quantization loss compared to the uniform quantizer, as shown in Figure 5. We hypothesize that L2 loss prioritizes the impact of larger values while neglecting smaller values, which is inappropriate given the unique distribution.

We conduct comparative experiments to further validate the contribution of minor values to the discriminability of temporal features. Specifically, in setting 1, we zero the interval $[-0.5min, 0.5max]$, and in setting 2, we apply a noise mask ranging from $[-1.5, 1.5]$ to scale values within the $[0.9max, max]$ interval, where $max$ and $min$ denote the maximum and the minimum for the temporal feature, respectively. As detailed in Table 4, the model exhibited robustness to disturbances in large values, while homogenization of small values lead to a collapse in model performance, underscoring the critical importance of minor values in preserving the discriminability of temporal features.

This analysis indicates the necessity of designing a quantizer specifically for the unique distribution of temporal features and

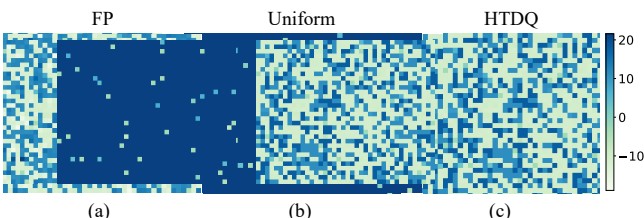

**Figure 2: Overview of QVD. The left is the High Temporal Discriminability Quantization, which uses the HiDi-TQ quantizer to retain the low TDScore of temporal features. The red arrow points to the location of the outlier. The right is the Scattered Channel Range Integration, which aims at mitigating the discreteness and asymmetry in inter-channel activation ranges, thereby enhancing the utilization rate of quantization levels by individual channels.**

developing a metric to measure the discriminability of temporal features. Driven by the above motivations, we propose the **T**emporal **D**iscriminability Score (**TDScore**) and **Hi**gh **Di**scriminability Quantizer for the Temporal Feature (**HiDi-TQ**).

*4.1.1 Temporal Discriminability Score.* The TDScore evaluates the similarity between the current temporal feature and several adjacent temporal features. We denote the discriminability score of the $t$-th temporal feature as $TDScore_t$. We initially apply a logarithmic function to $T_{emb}^t$ to enhance focus on minor values within the temporal feature as defined in Equation 9:

$$T_{emb}^{t'} = sign(T_{emb}^t) \cdot \left| log_2 \left| T_{emb}^t \right| \right|. \quad (9)$$

Subsequently, we compute the mean cosine similarity between the feature and its $n$ contiguous time steps following Equation 10:

$$TDScore_t = \frac{1}{n} \sum_{i=t+1}^{i=t+n} \cos\_sim(T_{emb}^{t'}, T_{emb}^{i'}). \quad (10)$$

A lower TDScore indicates higher discriminability of the temporal feature. We can employ the TDScore to assess the efficacy of quantization precisely.

*4.1.2 High Discriminability Quantizer for Temporal Feature.* As aforementioned, the quantization of temporal features not only necessitates minimizing quantization loss but also preserving distinctions between time steps. We evaluate quantizers calibrated using min-max or mean squared error (MSE) calibration strategy, as illustrated in Figure 5. These quantizers minimize quantization loss but concurrently diminish the distinguishability of temporal features

**Figure 3: Heat maps of full-precision temporal feature and its quantized versions of the uniform quantizer and the HTDQ.**

(TDScores are near 1). We introduce the high discriminability quantizer considering the unique distribution of temporal features. This quantizer is based on a logarithmic quantizer which is non-uniform. It allocates more quantization levels to values concentrated around zero, unlike uniform quantizers. Conversely, sparse distributions of large values are allocated fewer quantization levels. However, the vanilla logarithmic quantizer maps both the positive part and the negative part of temporal features to the same positive interval. This causes the data concentration regions of positive and negative intervals to overlap, exacerbating data concentration. Consequently, this reduces the utilization of available quantization levels. To address this issue, a relaxation coefficient $\beta$ is introduced, as follows:

$$T_{emb}^z = sign(T_{emb}) \cdot clip\left( \left\lfloor -log_2 \frac{|T_{emb} - \beta|}{s} \right\rfloor, 0, 2^b - 1 \right). \quad (11)$$

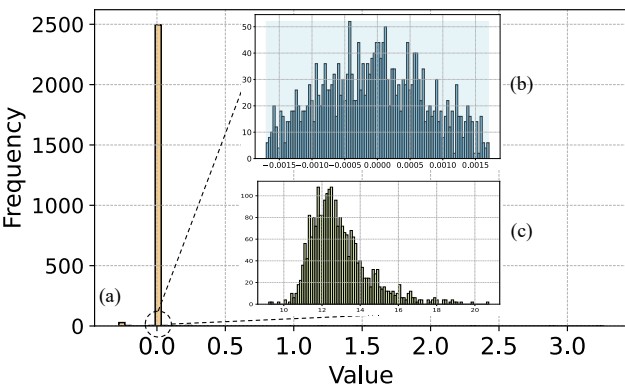

**Figure 4: Histogram of the temporal feature. (a) shows the histogram of 100% data of a temporal feature. (b) presents the middle 90% of the data which is concentrated within the range of $[-0.002, 0.002]$ and a single blue rectangle in the background indicates that these data are mapped to the same value. (c) depicts the distribution of the middle 90% of data processed through a logarithmic function, covering 10 quantization levels.**

The quantization process is initiated by a $\beta$ shift in the temporal features, thereby reducing the clustering of absolute values. $\mathbf{T}^z_{emb}$ represents the quantized temporal features. The scaling factor $s$ is typically set to be greater than the maximum absolute value of $\mathbf{T}_{emb}$ to ensure that the scaled activation values fall within the range $[0, 1]$. Additionally, $s$ can be adjusted to modify the data concentration, where increasing the value of $s$ narrows the range of activation values and makes it more concentrated.

As illustrated in Figure 5, although the logarithmic quantizer ensures foundational temporal discrimination (low TDScore), it also incurs a considerable L2 loss (MSE loss). To balance the precision of quantization and temporal discrimination, we utilize a composite metric $K$ as defined in Equation 12, which incorporates both TDScore and L2 loss, to search the optimal $s$ and $\beta$:

$$K = \sum_{i=1}^{T} TDScore_i + \sum_{i=1}^{T} (\mathbf{T}^i_{emb} - \hat{\mathbf{T}}^i_{emb})^2. \quad (12)$$

The TDScore allows $K$ to prioritize smaller values, while the L2 component ensures attention to losses in larger values. To mitigate truncation errors, We constrain the value of $s$ to the interval

$$\left[ max(|\mathbf{T}_{emb}|), \frac{min(|\mathbf{T}_{emb}|) + eps}{2^{1-2^n}} \right]. \quad (13)$$

This typically spans a very large range, to enhance search efficiency, we employ exponential step sizes where $s_i = max\,|\mathbf{T}_{emb}| \times 2^{0.05 \times i}$.

## 4.2 Scattered Channel Range Integration

Compared to the image diffusion models, we observe significant inter-channel variations in the video diffusion model. As depicted in Figure 1(b), we plot box plots for the activation sampled from the image diffusion model and output of the temporal attention block in the video diffusion model. It is evident that the activations generated by the temporal attention block exhibit discrete and asymmetric characteristics, referred to as inter-channel variations. The narrow range

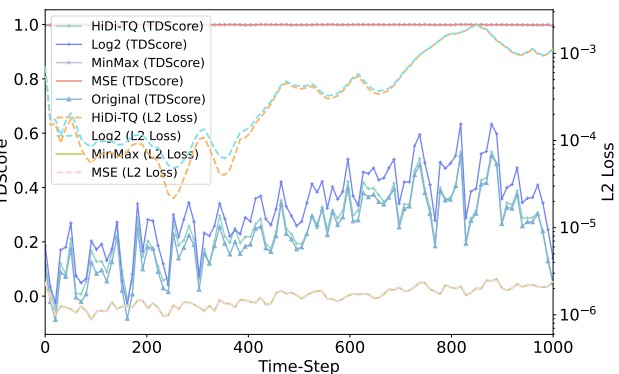

**Figure 5: Performance of various quantization strategies. The left axis represents the TDScore. The right axis delineates the L2 quantization loss.**

of individual channels leads to minimal overlap in the activation ranges across channels, resulting in low coverage of quantization levels per-channel, as illustrated in Figure 6, this diminishes the performance of the quantized model.

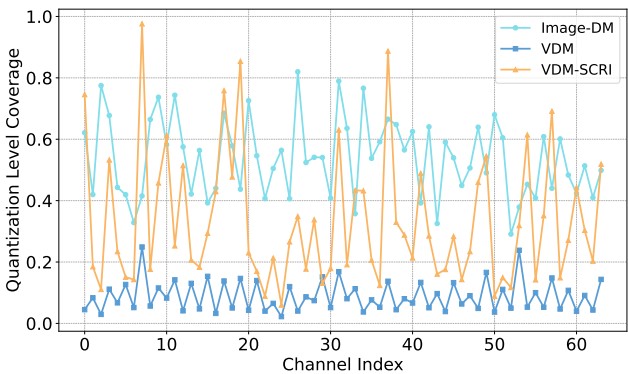

**Figure 6: Quantization level coverage of activation sampled from the image and video diffusion models. It is defined as the ratio of each channel's range to the overall range used for quantization. VDM-SCRI denotes the SCR-integrated version.**

Unlike the issues previously identified in large language models (LLMs), the problem of inter-channel variations is particularly pronounced in video diffusion models. This issue differs significantly from the outliers extensively studied in LLMs. In LLMs, as indicated by OS+ [57], outliers manifest as extreme shifts in specific channels. These shifts are consistent across samples, which allows for the straightforward identification and adjustment of an accurate shift amount to align the channel midpoints. Such a method is effectively utilized in LLMs, where techniques like OS+ [57] align and scale channel midpoints in the activation of the Layer-Norm layer through a channel-wise shift to address these outliers. However, in video diffusion models, the challenge of inter-channel variations requires distinct approaches due to its more complex and variable nature compared to the channel-specific shifts observed in LLMs. Specifically,

in video diffusion models, most channels exhibit varying degrees of shift that change with each sample, making it challenging to calculate an accurate shift for aligning the channel midpoints.

To address this issue, we propose the method of **S**cattered **C**hannel **R**ange **I**ntegration (**SCRI**). Formally, SCRI is a straightforward approach, as illustrated in Equation 14:

$$\widetilde{X} = X \oslash s, \quad s = \frac{X_{cmax}}{t}, \tag{14}$$

where $X$ denotes the activations with $C$ channels, while $s$, a vector of length $C$ similar to that used in OS+ [57], is employed to adjust the range of activation values. $X_{cmax}$ represents a vector composed of the maximum activation values within each channel, and $t$ serves as an adaptive parameter.

The SCRI, through meticulous design, can amplify the range of activation values for each channel, reducing discreteness and increasing overlap. However, an excessively large activation range is not conducive to improving the coverage of quantization levels for individual channels. To strike a balance between the activation range and the coverage of quantization levels, identifying the optimal value for $s$ is essential. Specifically, through a forward pass, we use a calibration set to determine the maximum activations values $X_{cmax}$. Concurrently, we determine the optimal $t$ using a grid search approach, confining our search within $[min(X_{cmax}), max(X_{cmax})]$. The optimization criterion during this search is the minimization of the MSE loss, comparing the output of the quantized model block against that of the full-precision model, as follows:

$$\arg\min_t \left\| \mathcal{F}(W, X) - \mathcal{F}_Q(W, X; t) \right\|, \tag{15}$$

where $\mathcal{F}$ represents the mapping function for the layers following LayerNorm in temporal attention modules and self-/cross-attention modules in the video diffusion model and $\mathcal{F}_Q$ denotes the mapping function corresponding quantized module.

Specifically, we identify several layers to apply SCRI: the Feed Forward Network (FFN), the projection layer, typically a linear layer or a convolution layer, and the attention layer subsequent to the LayerNorm. Similar to OS+ [57], we implement SCRI by making equivalent transformations to LayerNorm and subsequent layers, which incurs no additional overhead during inference. As depicted in Figure 6, SCRI significantly enhances the coverage of quantization levels by individual channels.

## 5 EXPERIMENTS

### 5.1 Implementation Details

**Datasets and Quantization Settings.** Video synthesis experiments are conducted on two cutting-edge models, MagicAnimate [60] and AnimateDiff [17], utilizing the TED-talks [48], FS-COCO [11] and COCO Captions [8] datasets. Both the input and output layers are consistently set at an 8-bit representation, whereas all remaining convolutional and linear layers are quantized in accordance with the predetermined target bit-width. To calibrate the models accurately, samples from all time steps of application (25 in this work) are collected to form a calibration set, corresponding to one video consisting of 16 consecutive frames. This process ensures that the models are finely tuned for generating high-quality video sequences, establishing a benchmark in the domain of video synthesis.

**Table 1: Quantization results on motion-guided video generation with TED-talks.** * denotes our implementation according to open-source codes.

| Method | Bits (W/A) | Size (Gb) | TBOPs | TED-Talks FID-VID↓ | FVD↓ |
|---|---|---|---|---|---|
| Full Precision | 32/32 | 22.8 | 9735 | 44.47 | 361.54 |
| Linear Quant* [43] | 8/8 | 5.7 | 716 | 80.70 | 618.33 |
| PTQ4DM* [47] | 8/8 | 5.7 | 716 | 77.55 | 590.89 |
| Q-Diffusion* [33] | 8/8 | 5.7 | 716 | 75.16 | 593.81 |
| **QVD** | 8/8 | 5.7 | 716 | **49.38** | **385.77** |
| Linear Quant* [43] | 6/8 | 4.3 | 555 | 82.43 | 649.02 |
| PTQ4DM* [47] | 6/8 | 4.3 | 555 | 84.40 | 644.42 |
| Q-Diffusion* [33] | 6/8 | 4.3 | 555 | 83.03 | 644.37 |
| **QVD** | 6/8 | 4.3 | 555 | **50.94** | **386.47** |
| Linear Quant* [43] | 6/6 | 4.3 | 430 | 119.02 | 1130.76 |
| PTQ4DM* [47] | 6/6 | 4.3 | 430 | 109.08 | 1074.93 |
| Q-Diffusion* [33] | 6/6 | 4.3 | 430 | 110.42 | 1006.69 |
| **QVD** | 6/6 | 4.3 | 430 | **77.54** | **683.72** |

**Evaluation Metrics.** In our experimental framework, we report the spatiotemporal fidelity of video synthesis using the FID-VID [1] and FVD [52] metrics, combining image quality assessment through Fréchet Inception Distance with temporal coherence evaluation using Fréchet Video Distance. Following AnimateDiff [18], semantic integrity is evaluated using the CLIP metric [45], which leverages a sophisticated language-image pretraining model to measure the alignment between generated animations and reference imagery. This involves calculating the cosine similarity between CLIP image embeddings of each animation frame and reference images, thereby evaluating domain similarity (Domain). Furthermore, our assessment is enhanced by examining the similarity between the prompt embeddings and individual frames to assess text alignment (Text) and by analyzing the similarity between consecutive frames to evaluate motion smoothness (Smooth). For computational performance, we calculate Bit Operations (BOPs) [3, 19] per video diffusion model during each forward pass, balancing efficiency with processing demands.

### 5.2 Main Results

**Motion Guided.** In this section, we apply our quantization method to the task of Human Image Animation, utilizing the MagicAnimate [60] framework on the TED-talks dataset [48]. MagicAnimate uses a motion sequence and a reference image to generate a video clip where the sequence directs the person's movements. We use the plain round-to-nearest Linear Quantization [43], PTQ4DM [47] and Q-Diffusion [33] as our baselines. As shown in table 1, our QVD consistently outperforms other methods by a large margin. Actually, these methods specifically designed for image diffusion, namely PTQ4DM and Q-Diffusion, do not achieve significant performance improvements in video diffusion quantization. On the contrary, at W8A8, our QVD gains a FID-VID reduction of 25.78 ( $75.16 \rightarrow 49.38$ compared with Q-Diffusion) and a FVD reduction of 205.12 ($590.89 \rightarrow 385.77$ compared with PTQ4DM) on TED-Talks, which is almost lossless. When the bit-width decreases (*i.e.*, W6A8 and W6A6), other methods drop drastically while our QVD maintains the performance to a great extent, merely introducing a 1.56 upswing in FID-VID and a 0.7 upswing in FVD.

**Table 2: Quantization results on motion-guided and sketch-guided video generation with FS-COCO and COCO Captions, respectively. * denotes our implementation according to open-source codes.**

| Method | Bits (W/A) | Size (Gb) | TBOPs | FS-COCO | | | COCO Captions | | |
|---|---|---|---|---|---|---|---|---|---|
| | | | | CLIP-Metric | | | CLIP-Metric | | |
| | | | | Text.↑ | Domain.↑ | Smooth.↑ | Text.↑ | Domain.↑ | Smooth.↑ |
| Full Precision | 32/32 | 22.8 | 9735 | 29.87 | 75.23 | 99.03 | 30.34 | 89.92 | 95.71 |
| Linear Quant* [43] | 8/8 | 5.7 | 716 | 28.05 | 69.40 | 98.68 | 30.00 | 84.97 | 92.80 |
| PTQ4DM* [47] | 8/8 | 5.7 | 716 | 28.23 | 70.92 | 98.46 | 29.61 | 83.88 | 91.49 |
| Q-Diffusion* [33] | 8/8 | 5.7 | 716 | 28.73 | 72.47 | 98.87 | 29.77 | 83.84 | 91.52 |
| **QVD** | 8/8 | 5.7 | 716 | **29.67** | **75.48** | **98.92** | **30.21** | **90.39** | **95.66** |
| Linear Quant* [43] | 6/8 | 4.3 | 555 | 28.19 | 69.59 | 98.77 | 29.93 | 83.80 | 91.92 |
| PTQ4DM* [47] | 6/8 | 4.3 | 555 | 28.96 | 71.39 | 98.48 | 29.93 | 83.13 | 90.62 |
| Q-Diffusion* [33] | 6/8 | 4.3 | 555 | 28.87 | 71.54 | 98.79 | 29.70 | 82.89 | 90.79 |
| **QVD** | 6/8 | 4.3 | 555 | **29.71** | **75.24** | **98.83** | **30.15** | **89.40** | **94.93** |
| Linear Quant* [43] | 6/6 | 4.3 | 430 | 26.32 | 64.52 | 96.35 | 29.24 | 78.51 | 91.91 |
| PTQ4DM* [47] | 6/6 | 4.3 | 430 | 27.83 | 68.43 | 96.12 | 27.98 | 72.83 | 89.58 |
| Q-Diffusion* [33] | 6/6 | 4.3 | 430 | 27.99 | 69.02 | 96.20 | 28.23 | 74.07 | 89.81 |
| **QVD** | 6/6 | 4.3 | 430 | **28.88** | **70.92** | **96.86** | **29.57** | **80.50** | **92.61** |

**Table 3: The effect of different methods proposed in the paper on TED-talks.**

| Method | Bits (W/A) | FID-VID↓ | FVD↓ |
|---|---|---|---|
| Full Precision | 32/32 | 44.47 | 361.54 |
| Linear Quant (Baseline) | 8/8 | 80.70 | 618.33 |
| + HTDQ | 8/8 | 61.22 | 483.99 |
| + SCRI | 8/8 | 67.57 | 529.70 |
| QVD (HTDQ + SCRI) | 8/8 | **49.38** | **385.77** |

**Table 4: Detailed ablation of temporal features. † denotes using the full-precision temporal feature.**

| Method | Bits (W/A) | FID-VID↓ | FVD↓ |
|---|---|---|---|
| Full Precision | 32/32 | 44.47 | 361.54 |
| SCRI + Linear Quant | 8/8 | 67.57 | 529.70 |
| SCRI + Log2 Quant | 8/8 | 50.43 | 398.07 |
| SCRI + HTDQ | 8/8 | 49.38 | 385.77 |
| Linear Quant † | 8/8 | 64.85 | 457.51 |
| Linear Quant | 8/8 | 80.70 | 618.33 |
| perturbates outliers | 8/8 | 65.94 | 487.91 |
| perturbates values near zero | 8/8 | 75.04 | 580.67 |

**Table 5: Detailed ablation of SCRI.**

| Method | Bits (W/A) | FID-VID↓ | FVD↓ |
|---|---|---|---|
| Full Precision | 32/32 | 44.47 | 361.54 |
| HTDQ + OS+ [57] | 8/8 | 51.21 | 407.93 |
| HTDQ + OS+ [57] without shift | 8/8 | 50.48 | 405.16 |
| HTDQ + SCRI | 8/8 | **49.38** | **385.77** |

**Table 6: Different calibration settings for SCRI.**

| Frame | Time-step | FID-VID↓ | FVD↓ | Time-cost(s) |
|---|---|---|---|---|
| 1 | 1 | 52.37 | 415.64 | 100.87 |
| 16 | 1 | 51.21 | 407.93 | 100.87 |
| 16 | 25 | 50.60 | 406.95 | 2534.51 |

**Image-guided and Sketch-guided** To demonstrate the superiority and versatility of our QVD, we further conducted experiments on AnimateDiff [18]. AnimateDiff utilizes both a visual modality (image or sketch) and a text prompt as inputs to generate videos. Our quantitative comparison mainly focuses on text alignment, domain similarity, and motion smoothness by employing the CLIP metric. Specifically, for the text alignment, our approach achieves a pioneering breakthrough on the FS-COCO dataset with Text-CLIP, surpassing 29 for the first time. For the domain similarity, Our QVD method leads the Q-Diffusion by 6.51 on the Domain-CLIP at W6A8 when evaluated on the COCO Captions dataset. For the motion smoothness, our method exhibits extraordinary video coherence on W8A8, remarkably achieving a score of only 0.05 lower than that of full precision (Smooth-CLIP).

## 5.3 Ablation Studies

In our ablation studies, we meticulously dissect the impact of each component in our QVD. These experiments are performed using the TED-talks dataset with W8A8 quantization, aiming to elucidate the individual contributions of different components to the overall effectiveness of the model. Linear quantization served as the baseline, adopting the MinMax calibration strategy for weight quantization and the MSE calibration strategy for activation quantization. Our analysis involves examining the effects of SCRI and HTDQ, as detailed in the following.

**Overall Effect of HTDQ and SCRI.** Table 3 outlines the collective impact of these components. The baseline model with linear quantization scores an FID-VID of 80.70 and an FVD of 618.33. Incorporating HTDQ remarkably improves these metrics to 61.22 and 483.99, respectively. Similarly, adding SCRI yields an improvement, achieving scores of 67.57 (FID-VID) and 529.70 (FVD). The

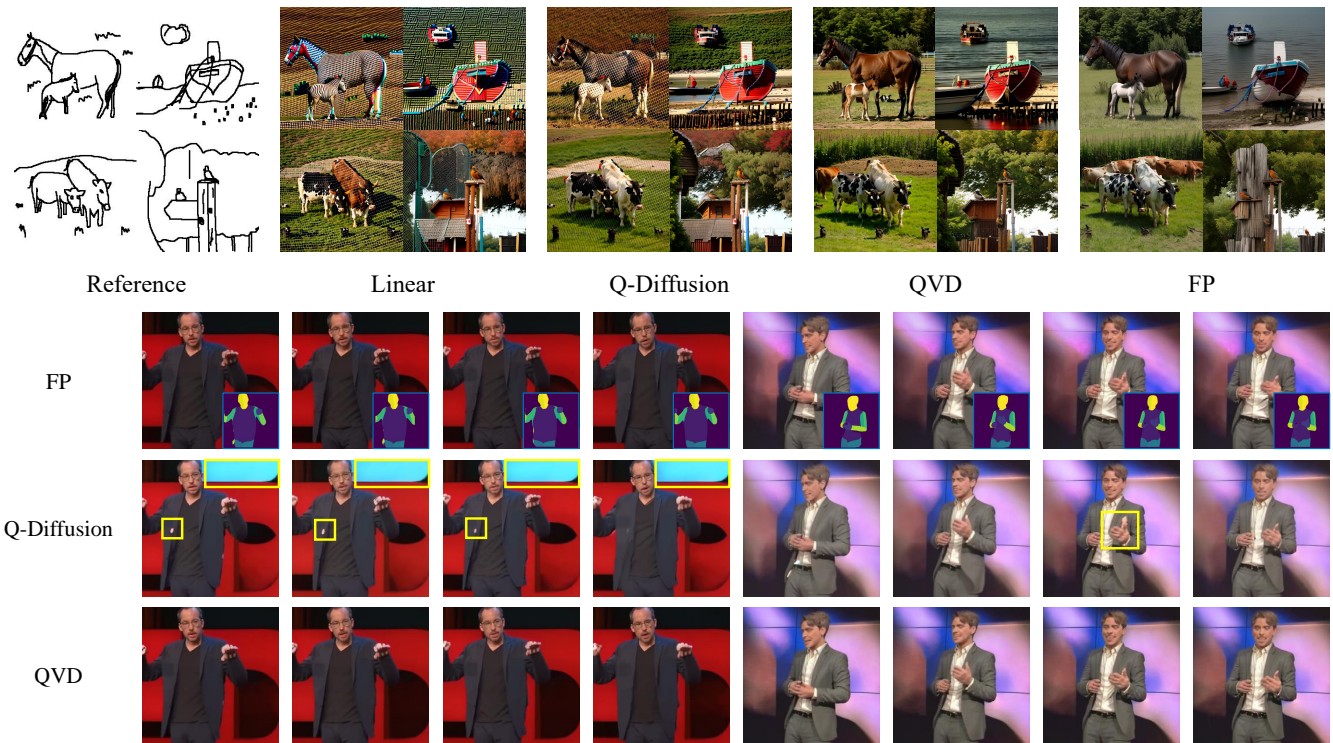

**Figure 7: A comparison of generation samples between Q-Diffusion, QVD, and the full-precision model. The upper figure demonstrates the generative outcomes of AnimateDiff, while the lower figure showcases the results from the Magic-Animate model.**

combination of both HTDQ and SCRI (our complete Magic-animate model) further reduces these scores to 49.38 (FID-VID) and 385.77 (FVD), underlining the benefit of integrating both components.

**Detailed Ablation of HTDQ.** To dissect the influence of HTDQ, we consider two stages in its integration (Table 4): the log quantization and our enhanced HTDQ. The experimental data presented in Table 4 clearly indicates the superiority of our enhanced HTDQ over the log quantization. When comparing the *SCRI+Log2 Quant* method and that with full *SCRI+HTDQ*, there is a noticeable improvement in both FID-VID and FVD scores, dropping from 50.43 to 49.38 for FID-VID, and from 398.07 to 385.77 for FVD. This demonstrates that our HTDQ, with its additional refinements, effectively enhances the quality of the quantized animations, offering a more faithful representation compared to the baseline.

**Exploring the Impact of SCRI.** In our ablation study detailed in Table 5, we compare the impact of different scaling methods on quantization quality. When the shift operation is removed from the OS+ [57] method, we see a modest improvement in both FID-VID (reduced from 51.21 to 50.48) and FVD (lowered from 407.93 to 405.16). Further refining the scale computation in our SCRI method yields even better results, with a notable decrease in FID-VID to 49.38 and FVD to 385.77, demonstrating the effectiveness of our adjustments in scale calculation for quantization.

**Investigating Calibration Efficiency.** To enhance the efficiency of the search process of SCRI, we investigate the impact of varying the number of frames and time steps on the calibration outcomes.

As shown in Table 6, a calibration setup using 16 frame features and 1 time-step achieves a balance between performance and time efficiency. All the experiments are conducted based on this setup.

## 5.4 Comparison of Visualization Results

Figure 7 shows the W8A8 qualitative results on FS-COCO and TED-talks datasets. When the reference is sketch (top), Linear Quantization and Q-Diffusion exhibit grid-like artifacts in their prediction results, while QVD successfully eliminates this issue and more closely approximates the performance of a full-precision model. When the reference is the motion video (bottom), Q-Diffusion generates noise and disordered backgrounds in its outputs. Moreover, QVD demonstrates superior detail retention for moving objects, exemplified by the hand of the speaker in the lower right corner.

## 6 CONCLUSION

In this work, we explore the application of post-training quantization in video diffusion models. We identify the significance of distinctive features for high-quality video generation and introduce the HTDQ method to preserve the discriminability of temporal features after quantization. Additionally, we observed a severe inter-channel variation issue in video diffusion models. To address this, we proposed the SCRI method to integrate activations across channels, thereby enhancing the performance of the quantized model. Our proposed QVD quantization framework is the first to quantize video diffusion models to 8-bit without significant performance degradation.

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
