# OpenReview forum: "QVD: Post-training Quantization for Video Diffusion Models"
_acmmm.org/ACMMM/2024/Conference — MM2024 Poster_

### Official Review · Reviewer_h82k · 2024-05-17

**Rating:** 5
**Confidence:** 2

**Summary:**

This paper investigates post-training quantization for video diffusion model. They design the first video-level quantization method, which consider high reliant on discriminable temporal features and inter-channel variations. Specifically, they introduce SCRI to improve the coverage of quantization levels across individual channels, TDScore to quantify temporal similarity, and the HiDi-TQ quantizer to keep high
discriminability of temporal features. Extensive experiments on various video diffusion models and datasets are conducted to demonstrate the effectiveness of the proposed method to improve the efficiency of VDMs while maintaining decent generation performance.

**Strengths:**

The motivation is greatly described. It is an interesting work to investigate pos-training quantization for video diffusion models.
This work is the first to conduct the quantization for VDM. It can be a plug-and-play process to accelerate any on-the-shelf VDMs.
They introduce TDScore to quantify temporal similarity and HiDi-TQ to keep high discriminability of temporal features, which are reasonable.

**Limitations:**

1. In Fig. 6, the relation between the three groups should be better explained
2. More visualization results from the Magic-Animate are appreciated because the improvement on the first case seems slight. Also, The abbreviation should be described in Fig. 7, e.g., 'FP'.
3. The ablation study on log2 quantization seems missing.

**Suitability:**

2

---

### Official Review · Reviewer_yuXU · 2024-05-21

**Rating:** 4
**Confidence:** 1

**Summary:**

This paper proposes QVD, the first post-training quantization scheme for video diffusion models. The paper introduces High Temporal Discriminability Quantization (HTDQ), which includes a High Discriminability Temporal Quantizer (HiDi-TQ) that prioritizes numerous near-zero values and maintains high temporal identifiability, as well as a Temporal Discriminability score (TDScore) to evaluate the similarity of adjacent temporal features. Additionally, the paper introduces the Scattered Channel Range Integration (SCRI) method, which enhances the coverage of quantization levels by individual channels through per-channel integration operations, thereby addressing the discrete range of activations.

**Strengths:**

(1) This paper proposes the QVD quantization framework, which is the first PTQ method tailored explicitly for video diffusion models.

(2) This paper identifies the critical inter-channel variations issue in video diffusion models and highlights the significance of accurate and discriminable temporal features for video generation.

(3) It introduces SCRI to improve the coverage of quantization levels across individual channels, TDScore to quantify temporal similarity, and the HiDi-TQ quantizer to maintain high discriminability of temporal features.

**Limitations:**

**Limited Generalizability**: The paper focuses mainly on video diffusion models, leaving its applicability to other types of video models or neural networks unverified.

**Implementation Complexity**: The QVD framework introduces multiple components, such as SCRI, TDScore, and HiDi-TQ, which could complicate implementation and integration into existing systems.

**Suitability:**

2

---

### Official Review · Reviewer_pCni · 2024-05-24

**Rating:** 4
**Confidence:** 3

**Summary:**

The authors propose a novel method for post-training quantization (PTQ) of video diffusion models. They first explore failures of existing PTQ and identify two main concerns, namely quantization of temporal feature activations into mostly one bin, and significant inter-channel variation of activations which leads to many bins having no quantized values. The authors then propose methods to resolve these issues, namely a novel quantizing strategy HiDi-TQ that spreads activations into more bins, which is calibrated according to a novel TDScore (which measures temporal discriminability), and SCRI, which scales activations to use a wider range of quantization bins between channels. Their proposed method is extensively evaluated and they show highest performance in generation compared to other PTQ methods.

**Strengths:**

* Insightful experiments to understand the drawbacks of PTQ on video diffusion models
* Creative and novel solutions to these issues, via the TDScore, HDTQ, and SCRI
* Extensive evaluations
* Good performance

**Limitations:**

* I still have a bit of confusion on the two drawbacks. It seems both lead to only a few quantization bins having values, and I find that the motivations/reasonings for the proposed method (specifically SCRI) are only explained at a high level. I would appreciate a deeper explanation, although I am not so familiar with LLM quantization which may be the reason for my confusion.
* Are HiDi-TQ, HTDQ, and high discriminability quantizer for temporal feature all the same thing? It should be given one name or explained more clearly if they are different.
* Fig. 1 is confusing, it would be better with labels and I think it could be laid out more clearly.

**Suitability:**

3

---

### Official Review · Reviewer_kpu5 · 2024-05-26

**Rating:** 3
**Confidence:** 1

**Summary:**

This paper presents QVD, a pioneering post-training quantization approach tailored for video diffusion models. It introduces two key methods: High Temporal Discriminability Quantization (HTDQ) for maintaining temporal features and Scattered Channel Range Integration (SCRI) to enhance quantization level coverage across channels.

**Strengths:**

1. **Innovativeness**: The paper proposes QVD, the first quantization framework specifically tailored for video diffusion models, afilling a notable gap in current research.
2. **Clear Structure**: The paper is well-structured and clearly written, making it easy to follow and understand the proposed methods and results.

**Limitations:**

**(Major)** As a work related to video generation, I did not see any video materials in the supplementary content.

**(Minor)** Given the complexity of the QVD, it is unclear whether it can serve as a convenient, plug-and-play plugin for existing video diffusion models, potentially limiting its ease of adoption and practical utility.

**Recommendations**:

Include Video Materials: To better demonstrate the effectiveness and practical utility of the proposed methods, it is essential to include relevant video materials. I recommend providing comparisons of the quantized and original models across different video diffusion methods, as well as comparisons with previous approaches. This is necessary because static images can be cherry-picked, making it difficult to assess the true quality of the video.

Conduct Portability and Generalizability Experiments: Consider conducting experiments that demonstrate the portability and generalizability of QVD across more video diffusion models and datasets. This would help in understanding the adaptability of QVD in diverse settings. Given the workload, it is acceptable not to provide comparisons with previous methods, but it is still necessary to show comparisons between the quantized and original models.

**Suitability:**

2

---

### Meta-Review · Area_Chair_H6uY · 2024-07-01

**Recommendation:** Accept (Poster)
**Confidence:** 5

**Metareview:**

This work presents an approach to post-training quantization for video diffusion models, with nearly no degradation under the w8a8 setting. The paper is well-written, and the approach is innovative. The rebuttal has addressed the concerns of all the reviewers. After the rebuttal, the reviewers are positive about the work. Thus the AC recommendation acceptance of this work.